# Relevance of Routine Postoperative CT Scans Following Aneurysm Clipping—A Retrospective Multicenter Analysis of 423 Cases

**DOI:** 10.3390/jcm11237082

**Published:** 2022-11-29

**Authors:** Michał Krakowiak, Justyna Małgorzata Fercho, Tomasz Szmuda, Kaja Piwowska, Aleksander Och, Karol Sawicki, Kamil Krystkiewicz, Dorota Modliborska, Sara Kierońska, Waldemar Och, Zenon Dionizy Mariak, Jacek Furtak, Stanisław Gałązka, Paweł Sokal, Paweł Słoniewski

**Affiliations:** 1Department of Neurosurgery, Medical University of Gdansk, 80-210 Gdansk, Poland; 2Student’s Scientific Circle of Neurosurgery, Neurosurgery Department, Medical University of Gdansk, 80-952 Gdansk, Poland; 3Department of Neurosurgery, Provincial Hospital in Olsztyn, Niepodległości 44, 10-045 Olsztyn, Poland; 4Department of Neurosurgery, Medical University in Białystok, Jana Kilińskiego 1, 15-089 Białystok, Poland; 5Department of Neurosurgery and Neurooncology, Nicolaus Copernicus Memorial Hospital, 93-513 Lodz, Poland; 6Department of Neurosurgery, Provincial Specialist Hospital in Słupsk, Hubalczyków 1, 76-200 Słupsk, Poland; 7Department of Neurosurgery and Neurology, Jan Biziel University Hospital Nr 2 Collegium Medicum, Nicolaus Copernicus University, 85-168 Bydgoszcz, Poland

**Keywords:** computed tomography, intracranial aneurysm, postoperative care

## Abstract

Aim: Postoperative head computed tomography (POCT) is routinely performed in numerous medical institutions, mainly to identify possible postsurgical complications. This study sought to assess the clinical appropriateness of POCT in asymptomatic and symptomatic patients after ruptured or unruptured aneurysm clipping. Methods: This is a retrospective multicenter study involving microsurgical procedures of ruptured (RA) and unruptured intracranial aneurysm (UA) surgeries performed in the Centers associated with the Pomeranian Department of the Polish Society of Neurosurgeons. A database of surgical procedures of intracranial aneurysms from 2017 to 2020 was created. Only patients after a CT scan within 24 h were included. Results: A total of 423 cases met the inclusion criteria for the analysis. Age was the only significant factor associated with postoperative blood occurrence on POCT. A total of 37 (8.75%) cases of deterioration within 24 h with urgent POCT were noted, 3 (8.1%) required recraniotomy. The highest number necessary to predict (NNP) one recraniotomy based on patient deterioration was 50 in the RA group. Conclusion: We do not recommend POCTs in asymptomatic patients after planned clipping. New symptom onset requires radiological evaluation. Simultaneous practice of POCT after ruptured aneurysm treatment within 24 h is recommended.

## 1. Introduction

Postoperative head computed tomography (POCT) after aneurysm clipping surgery is a common practice. It is used mainly to identify potential complications, for the surgeons’ comfort and medico-legal aspects. Hematoma detection after craniotomies is one of the main reasons of urgent return to the OR [1,2].

As a potential life-threating condition, hematomas need to be detected as soon as possible and surgically evacuated if necessary. The wide availability of CT scanners has led to their routine use after intracranial surgeries. High costs, transport-related risk, personnel burden, and unnecessary radiation exposure are the main drawbacks of POCT. This retrospective, multicenter, observational study was aimed to evaluate the clinical utility of POCT in asymptomatic and symptomatic patients after aneurysm clipping. 

## 2. Materials and Methods

This is a retrospective multicenter study involving intracranial microsurgical procedures of ruptured (RA) and unruptured aneurysms (UA) performed in Centers associated with the Pomeranian Department of the Polish Society of Neurosurgeons. A database of surgical aneurysm clipping procedures from 2017 to 2020 was created. Only patients with a CT study (<24 h) were included. Medical records were reviewed retrospectively for gender, age, aneurysm location, and presence of subarachnoid hemorrhage. Due to the retrospective nature of the study, available information on patients taking anticoagulation/antiplatelet drugs was insufficient and therefore not included. If scheduled for surgery, preoperative laboratory assessment of the coagulation pathway and platelet function was routinely assessed and corrected if abnormal (PT ≤ 13.5 seconds or INR ≤ 1.4). In the RA group, modified Fisher and Hunt-Hess scale were gathered [3,4]. The first CT status (routine/urgent), hemorrhage detection and revision surgery were noted. If more than one aneurysm was clipped during surgery, the patient was assigned to multiple aneurysm groups with the symptomatic as primary.

The following patients were excluded from the study:-No CT within 24 h;-Admitted only for arteriography;-Provided with intravascular embolization;-Without qualifications or consent to the procedure;-Patients requiring craniectomy.

Deterioration was defined as failure to adequately awaken from general anesthesia, consciousness deterioration (Glasgow Coma Scale reduced by ≥2), abnormal severe headaches, new focal neurological signs, and seizures or signs of intracranial hypertension (pupillary dilatation, asymmetric pupillary reaction to light, decerebrate or decorticate posturing). A positive POCT included any unexpected findings on imaging that could not be interpreted as routine postoperative changes. Postoperative blood under the bone flap of thickness ≥ 5 mm was defined as postoperative hematoma.

### Statistical Analysis

PQStat v.1.8.0 (PQStat Software, Poznań, Poland) was used for statistical analysis [5]. Data was verified with the Shapiro–Wilk test and subsequently analyzed according to the test results. Descriptive statistics included arithmetic mean, median and standard deviation (std). Spearman’s rank correlation coefficient, Mann–Whitney U, χ^2^, and Fisher’s exact tests were used, as appropriate. Graphical illustrations were performed in Microsoft 365 (Microsoft Corporation ver.18.2008, Redmond, Washington, DC, USA).

## 3. Results

A total of 496 patients admitted to the departments of the involved centers were found. After exclusion, 423 cases were included into the database. The patients consisted of 70.92% women (*n* = 300) and 29.08% men (*n* = 123). Ruptured aneurysms accounted for 47.75% of cases (*n* = 202) and unruptured for 221 cases. The mean age of all patients was 57 (std = 12.9). There was statistical difference in the age between groups, indicating that RA were younger (*p* = 0.019). Within the ruptured and unruptured cohorts, female predominance was statistically higher in UA (*p* < 0.000014). The groups are compared in Table 1.

In RA, the mean Hunt Hess grade was 2 and the modified Fisher was mean 3. Aneurysm distribution is presented in Table 2. A total of 97.4% of aneurysms were located in the anterior circulation. Multiple aneurysms were clipped in 30 cases. 

### 3.1. POCT Findings

The number of next-day routine POCT and due to neurological deterioration in the RA and UA cohorts is presented in Table 3. Blood under the bone flap was seen in 151 = 35.7% POCTs with 87 = 39.4% in UA and 64 = 31.7% in the RA group. Intracerebral hemorrhage occurred in *n* = 10 scans (2.4%) and hematoma was present in *n* = 90 cases (21.3%).

### 3.2. Risk Factors for Postoperative Blood Occurrence

Age was the only significant factor associated with postoperative blood occurrence on POCT, with *p* = 0.043 (median age: 55.5 and 59). The monotonic relationship of blood thickness and age achieved r = 0.39 (CI: 0.24–0.52) and *p* = 0.000001. Sex, craniotomy site, and SAH presented no association (*p* > 0.05). The severity of modified Fisher scale, nor Hunt Hess grade occurred significant when analyzed with postoperative blood thickness or hematoma occurrence (*p* > 0.05). Multiple aneurysm clipping did increase the postoperative amount of blood *p* = 0.03, but did not increase hematoma occurrence: *p* = 0.65.

### 3.3. Deterioration and Recraniotomy

A total of 37 (8.75%) patients deteriorated within 24 h and required urgent POCT. There was no difference between RA and UA groups in the number of deteriorated patients (*p* > 0.05). Out of 37 deteriorations, 3 (8.1%) patients required recraniotomy, 1 in the RA, and 2 in the UA cohorts. During the first 24 h, 11 (2.6%) recraniotomies were performed due to acute hematoma. In the asymptomatic group, 6 (3.27%) RA and 2 (0.99%) UA patients required hematoma evacuation after routine POCT. The data are presented in Figure 1.

Sensitivity (S), specificity (E), positive predictive value (PPV), negative predictive value (NPV), predictive summary index (PSI), and number necessary to predict (NNP) were calculated both for symptomatic patients and positive POCT findings as diagnostic tools for recraniotomy [6]. 

The S = 0.14 of symptoms and PPV = 0.05 were the lowest in the RA group resulting in the highest NNP of 50 patients for 1 recraniotomy. Symptom occurrence was specific within the three groups, reaching a mean of 0.92. The full data are presented in Table 4.

The appliance of a positive POCT result as the diagnostic test had S of 1 in all groups. The highest E = 0.83 occurred in symptomatic RA patients. An altered POCT reached the lowest values of NNP for deteriorated UA and RA, of 3 and 4, respectively. Full data are presented in Table 5.

## 4. Discussion

After neurosurgical procedures, an early POCT is performed to detect complications such as hemorrhage, ischemia, oedema, or hydrocephalus [7]. Strict concomitant clinical observation with neurological examination is a standard in postsurgical care [8]. With radiological diagnostics the surgeon gains assessment of the postoperative field. On the other hand, abnormal postoperative findings are common; they do not directly lead to neurological deterioration, but tempt the surgeon to “fix the ct” [9,10].

Postoperative hematoma can be defined as bleeding under the bone flap at the site after surgery. In most cases, a small amount of blood is to be expected. This is why no clear definition of a postoperative hematoma was established [11]. With no definition, the presented occurrence varies. Considering aneurysm surgery in particular, it ranges from 4.9% to 47%, with a revision needed in 0.3–2.6% of the cases [12,13,14]. 

Contemporaneously medical imaging is one of the major sources of radiation exposure [15]. It has been evaluated that mean exposure during hospitalization in aneurysm coiling was 5.68 Gy in comparison to 1.77 Gy with clipping [16]. Although abdominal CT scans show higher risks than cranial CTs, 0.4% of all cancers in USA are caused by CT examinations [17]. Therefore, additional unnecessary radiological examination should be prevented. Recent literature indicates that routine POCT exhibits little clinical utility. Attempts were made to construct screening guidelines or algorithms for postcraniotomy radiological evaluation. None has come to a standard of care and wide clinical practice, leading to discrepancies in management between hospitals [7,10,18]. 

Even during transport to perform a POCT, the patient is susceptible of deterioration [19]. The advantage of solid neurological monitoring over postoperative imaging as a method of patient follow up has been recently reported in several other studies. In recent years, the value of POCT was questioned, showing its uselessness with no neurological deficits [10,18].

Zygourakis et al. examined the cost and clinical utility of postoperative CT among 304 patients after elective aneurysm clipping. The findings influenced clinical management in 3.6% of the cases, with only one patient requiring reoperation for an asymptomatic epidural hematoma. In the neurologically intact group, imaging altered management only in 1.1% of the cases, whereas in patients presenting non-focal and focal deficits treatment it changed in 4.8% and 9.0% of the cases, respectively [20]. Another study showed that among all the patients who underwent routine POCT, there were none noted with serious hematoma who would not have been identified by clinical manifestation [18]. However, there is still little consensus considering the exact timing of examination, which varies between studies [13,21,22,23].

Close neurological follow up can indeed detect potential bleeding yet requires symptom development to alert the medical team. Initially, the “four-hour rule” was established for subdural hematomas on the basis that earlier surgery provided better results [24]. Although conflicting evidence can be found, the timing of surgical intervention, may influence the final outcome [25,26].

In our series, the revision rate was higher than in Fukamachi’s, where 4 after 190 aneurysmal postoperative hematomas were operated (2.1%) [13]. However, in the ICH group (8.02%), our group achieved an ICH of 2.4%, showing lower results [27].

Moreover, economic aspects also need to be considered. During the COVID pandemic, various microsurgical procedures decreased due to the Polish healthcare transformation and social anxiety [28]. In Poland, in 2020 alone, 1057 unruptured aneurysms were treated. The clipping rate was 22.65% [29]. The price of one POCT in the University Clinical Centre in Gdansk scan is PLN 220 or USD 46.51 (using the exchange rate from 31 October 2022) [30]. Considering at least 1 POCT in every patient, the annual cost of this procedure fluctuates around USD 10,232.2 countrywide. In the presented study, unnecessary costs reached USD 9348.51 of the UA group. Real costs are hard to estimate due to various personnel and equipment involved in safe transport providence. 

It is worth mentioning that 2.07% of asymptomatic patients after POCT required revision surgery. Asymptomatic patients appeared in the series presented by Wen et al., where, out of 1.9 % (15/786) requiring recraniotomy due to hematoma, 0.7% (5/721) were without clinical manifestations [31]. Similar results were presented in a systematic review by Blumrich et al. In their paper, 0.756% of asymptomatic patients after POCT required intervention, with 21.4% on that basis qualifying for emergency operation [7]. In the work of Fontes and colleagues, approximately 90% of patients with no deterioration and expected deterioration had predictable POCTs. This resulted in a longer ICU stay, another radiological evaluation, but without active intervention [9]. In the authors opinion, the crucial question is when the patient can be safely transferred from an ICU to the neurosurgical ward without radiological examination.

### 4.1. Limitations

This is a retrospective study of patients operated within the Pomeranian Department of the Polish Society of Neurosurgeons. Moreover, not all potential risk factor data presented by other authors were available in the patient’s medical history [31,32].

### 4.2. Further Work

In order to maximize patient safety, an algorithm should be developed to decide if a prolonged ICU stay is needed to fully exclude any complications.

## 5. Conclusions

The clinical appropriateness of POCT in asymptomatic patients after planned clipping is doubtful, and it is therefore not recommended. If postsurgical deterioration occurs, urgent POCT is necessary. In patients after ruptured aneurysm surgery, POCT within 24 h is to be highly recommended.

## Figures and Tables

**Figure 1 jcm-11-07082-f001:**
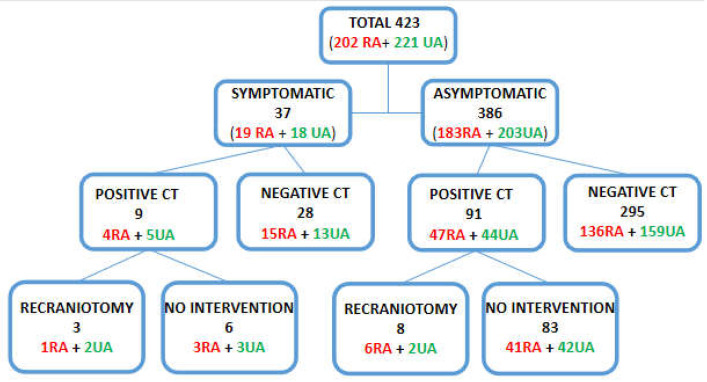
Division of patients into subgroups according to the occurrence of symptoms and POCT status (black color: all aneurysms; red color: ruptured aneurysms; green color: unruptured aneurysms).

**Table 1 jcm-11-07082-t001:** Comparison of age in the ruptured and unruptured aneurysms.

	Ruptured	Uunruptured
Count (Female/Male)	202 (123/79)	221 (177/44)
Mean	54 (std = 13.96)	57 (std = 11.75)
Median	53	58

**Table 2 jcm-11-07082-t002:** Aneurysms by location and count.

Aneurysm Location	Count [*n*]	Percentage [%]
ACOA (M)	9	2.128%
ACOA, ACA (Complex)	121	28.605%
BA	5	1.182%
BA (M)	2	0.473%
ICA	82	19.385%
ICA (M)	8	1.891%
MCA	180	42.553%
MCA (M)	10	2.364%
PCOA	1	0.236%
PCOA (M)	1	0.236%
PICA	3	0.709%
VA	1	0.236%

Abbreviations: ACOA: Anterior Communicating Artery; ACA: Anterior Cerebral Artery; BA: Basilar Artery; ICA: Internal Carotid Artery; MCA: Middle Cerebral Artery; PCOA: Posterior Communicating Artery; PICA: Posterior Inferior Cerebellar Artery; VA: Vertebral Artery; (M): multiple aneurysms clipped.

**Table 3 jcm-11-07082-t003:** Postoperative head computed tomography characteristics.

Patients	Count
Routine	386 (91.253%)
Deteriorated	37 (8.747%)
Ruptured aneurysms	19 (9.406%)
Unruptured aneurysms	18 (8.145%)

**Table 4 jcm-11-07082-t004:** Appliance of symptoms as a diagnostic test for recraniotomy.

Group	Test (Symptoms)	Recraniotomy	No Change	Sensitivity	Specificity	PPV	NPV	PSI	NNP
All	S+	3	34	0.27	0.92	0.08	0.98	0.06	16.67
	S−	8	378						
Ruptured	S+	1	18	0.14	0.91	0.05	0.97	0.02	50
	S−	6	177						
Uunruptured	S+	2	16	0.50	0.93	0.11	0.99	0.1	10
	S−	2	201						

Abbreviations: S+: symptomatic, S−: asymptomatic, PPV: positive predictive value, NPV: negative predictive value, PSI: predictive summary index, NNP: number necessary to predict.

**Table 5 jcm-11-07082-t005:** Appliance of postoperative head computed tomography result as a diagnostic test for recraniotomy.

Group	Test (POCT)	Recraniotomy	No Change	Sensitivity	Specificity	PPV	NPV	PSI	NNP
RA S+	POCT+	1	3	1	0.83	0.25	1	0.25	4.00
	POCT−	0	15						
RA S−	POCT+	6	41	1	0.77	0.13	1	0.13	7.69
	POCT−	0	136						
UA S+	POCT+	2	3	1	0.81	0.40	1	0.40	2.50
	POCT−	0	13						
UA S−	POCT+	2	42	1	0.79	0.05	1	0.05	20
	POCT−	0	159						

Abbreviations: RA: ruptured aneurysms, UA: unruptured aneurysms, S+: symptomatic, S−: asymptomatic, POCT+: scan positive, POCT−: scan negative, PPV: positive predictive value, NPV: negative predictive value, PSI: predictive summary index, NNP: number necessary to predict.

## Data Availability

Data available on request.

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
