# Peer review of "Relevance of Routine Postoperative CT Scans Following Aneurysm Clipping—A Retrospective Multicenter Analysis of 423 Cases"

_jcm, 2022, doi:10.3390/jcm11237082_

Round 1
Reviewer 1 Report
Whether routine cranial CT scans are necessary after craniotomy has been controversial in recent years. This study seeks to address this issue using CT scans with 24 h after aneurysm clipping as an entry point. By collecting data of 423 aneurysm clipping operations (containing both RAs and UAs) that underwent POCT within 24h at their unit from 2017-2020, the authors concluded through a series of appropriate statistical analyses that POCT is unnecessary in asymptomatic planned clipping and highly recommended in RA clipping. This is a valuable guide for managing patients with aneurysm clipping. However, there is still some room for improvement in this study.
1. Did all cases enrolled in the study exclude bleeding or coagulation-related disorders/history f medication use? The authors should have indicated this in the exclusion criteria.
2. Is there a gender difference between the RA and UA groups?
3. A subgroup analysis of patients in the RA group based on the hunt-hess score is advisable to assess whether POCT is recommended for all RA patients.
4. Please standardize the reference format (e.g., #20).
Author Response
On behalf of all the authors, I would like to thank the Reviewer for the encouraging feedback and careful review of our study. We have carefully considered the comments and would like to respond to them point-by-point as follows. All the changes in the manuscript are highlighted. By attending to many helpful suggestions, we believe that our manuscript has been improved significantly.
- Did all cases enrolled in the study exclude bleeding or coagulation-related disorders/history f medication use? The authors should have indicated this in the exclusion criteria.
A1. Information added in lines 49 - 53
- Is there a gender difference between the RA and UA groups?
A2. Information added in lines 85-86
- A subgroup analysis of patients in the RA group based on the hunt-hess score is advisable to assess whether POCT is recommended for all RA patients.
A3. Information added in lines 104-106
- Please standardize the reference format (e.g., #20).
A4. References have been corrected.
Kind regards,
Author
Reviewer 2 Report
Dear editor,
I read this article with great interest. The role of postoperative computed tomography (POCT) following cranial brain surgery has been repeatedly questioned over the past few years, but no consensus has been reached in practice. This article suggests that deprivation of POCTs in asymptomatic planned clipping and ruptured aneurysm surgery within 24 hours of POCTs is highly recommended.However, there are several considerations to be made before further consideration of this manuscript.
1. Summary Conclusion
The writer uses a long and unintelligible phrase. Suggest changes.
2. Methods The author only described the inclusion and exclusion criteria, but did not describe the statistical methods, and the corresponding description of the outcome evaluation criteria was missing, please add.
3. The author made a predictive assessment of postoperative bleeding status, but the corresponding data is missing, please add.
4. It is recommended that the author improve the subscripts of pictures and tables.
5. The author should indicate the position of Table 4 in the manuscript.
6. The authors' conclusion that POCT is recommended for deprivation in asymptomatic patients is inaccurate. The value of POCT in asymptomatic patients with unruptured aneurysms is relatively low. It is recommended that you make a clear statement and redescribe Result 3.3 based on the result.
Author Response
On behalf of all the authors, I would like to thank the Reviewer for the encouraging feedback and careful review of our study. We have carefully considered the comments and would like to respond to them point-by-point as follows. All the changes in the manuscript are highlighted. By attending to many helpful suggestions, we believe that our manuscript has been improved significantly.
- Summary Conclusion. The writer uses a long and unintelligible phrase. Suggest changes.
A1. The conclusion of the study has been corrected and presents clear recommendations based on the study data.
- Methods The author only described the inclusion and exclusion criteria, but did not describe the statistical methods, and the corresponding description of the outcome evaluation criteria was missing, please add.
A2. Paragraph 2.1 was added to the manuscript describing statistical methods used in the study.
- The author made a predictive assessment of postoperative bleeding status, but the corresponding data is missing, please add.
A3. Information has been included in lines 95 - 105
- It is recommended that the author improve the subscripts of pictures and tables.
A4. Figure and table subscriptions have been rewritten and corrected.
- The author should indicate the position of Table 4 in the manuscript.
A5. Position of Table 4 added.
- The authors' conclusion that POCT is recommended for deprivation in asymptomatic patients is inaccurate. The value of POCT in asymptomatic patients with unruptured aneurysms is relatively low. It is recommended that you make a clear statement and redescribe Result 3.3 based on the result.
A6. Conclusions have been rewritten to present a clear statement.
Kind regards,
Author
Round 2
Reviewer 2 Report
Please keep the results in the abstract consistent with those at the end of the paper. POCT also has relatively good detection results for postoperative symptomatic unruptured aneurysms, NNP:2.5. However, you concluded that POCT is not recommended for unruptured aneurysms. This is not reflected in the discussion and does not match your results. Please explain the situation.
Author Response
Dear Reviewer,
We thank for the effort put into the evaluation of our revised work. The manuscript was corrected adequately to the reply and all changes in the abstract and conclusions are marked red. NNP of 2.5 (Table 5) applies to patients with unruptured symptomatic aneurysms with a positive CT scan. In the UA group, symptoms alone resulted in an NNP=10 (Table 4). We did not include the necessity of urgent POCT in cases of postsurgical deterioration n the previous manuscript because such diagnostic management is a clinically justified routine practice. We hope that the second revision will fulfill You requirements.
Kind regards,
Author